# Patient Acceptability of the First Integrative Pediatric Oncology Unit in Spain—The Pediatric Cancer Center Barcelona Experience: A Retrospective Study

**DOI:** 10.3390/cancers17020222

**Published:** 2025-01-11

**Authors:** Esther Martínez García, Cristina López de San Roman Fernández, M. Betina Nishishinya Aquino, Sara Pérez-Jaume, Carles Fernández-Jané, Ofelia Cruz Martínez, Andrés Morales La Madrid

**Affiliations:** 1Integrative Pediatric Oncology Unit, Pediatric Cancer Center Barcelona, Sant Joan de Déu Barcelona Children’s Hospital, 08950 Barcelona, Spain; esther.martinez@sjd.es (E.M.G.); c.lopezsanroman@sjd.es (C.L.d.S.R.F.); mbetina.nishishinya@sjd.es (M.B.N.A.); 2Institut de Recerca Sant Joan de Déu, Esplugues de Llobregat, 08950 Barcelona, Spain; 3Health Department, Tecnocampus, Universitat Pompeu Fabra, Mataró-Maresme, 08302 Barcelona, Spain; 4Pediatric Neuro-Oncology, Pediatric Cancer Center Barcelona, Hospital Sant Joan de Déu, 08950 Barcelona, Spain; ofelia.cruz@sjd.es (O.C.M.); andres.morales@sjd.es (A.M.L.M.)

**Keywords:** acupuncture, children, adolescent, oncology, integrative oncology, pediatrics, cancer, pediatric oncology

## Abstract

This study explored the introduction of a new approach for treating children with cancer at the Pediatric Cancer Center in Barcelona, Spain. Traditionally, cancer treatment focuses on conventional approaches including surgery, chemotherapy, and radiation, but a significant percentage of families seek additional therapies, such as acupuncture and aromatherapy. This study aims to assess how well these complementary therapies are accepted by patients and their families. Over two years, the center offered these therapies to 433 patients, with the majority of families welcoming additional treatments. Very few rejected them, demonstrating that integrating these therapies into pediatric cancer care is feasible and well-received. The findings suggest that such therapies could play a valuable role in supporting young cancer patients, potentially encouraging more healthcare centers to adopt similar practices.

## 1. Introduction

Childhood developmental cancer refers to a group of diseases that arise during the formation of organs and tissues as part of the human developmental process. It can affect infants, children, adolescents, and young adults [1,2]. In most cases, the causes of these cancers remain unknown without obvious predisposition factors, although they may be related mostly to epigenetic factors during cellular development [3,4]. These potential lethal conditions during the first decades of life can have a strong physical and emotional impact on the patient and their family [5]. To cope with these challenges, many patients and families turn to complementary therapies, particularly when conventional allopathic treatments fail to provide sufficient relief. Complementary therapies are defined as medical practices and products not typically part of conventional medical care but used alongside standard treatments to improve patient well-being and quality of life [6]. Studies indicate that between 31 and 84% of families of oncologic patients combine conventional allopathic medicine with complementary therapies such as acupuncture, massage, meditation, herbs, homeopathy, and dietary supplements [7,8,9].

The World Health Organization (WHO), in its Traditional Medicine Strategy 2014–2023, highlights that the increased interest in traditional and complementary therapies demands its closer integration into healthcare systems [10]. The strategy aims to support Member States in harnessing the potential contribution of complementary therapies to health, well-being, and people-centered healthcare, and to promote the safe and effective use of complementary therapies through the regulation of products, practices, and professionals. This emphasis aligns with a growing acknowledgment of the importance of holistic approaches in addressing not only the physical aspects of diseases but also their emotional, social, and psychological dimensions, particularly in pediatric oncology, where patient vulnerability is compounded by developmental and familial factors [11,12,13,14]. The overall use across countries of these complementary treatments has increased in the last 30 years, revealing a real social demand [15,16,17]. In a recent study conducted in Germany, 40% of interviewed parents reported using complementary treatments and wished for their incorporation into the public health system, even if that would imply the individual coverage of its cost [18].

Integrative medicine is the combination of complementary therapies with conventional medical treatments in a coordinated manner, focusing on a holistic, patient-centered approach to care [6]. In the context of cancer care, integrative oncology extends this concept by incorporating mind and body practices, natural products, and lifestyle modifications with conventional cancer treatments [19,20]. The goal is to optimize health and quality of life across the cancer care continuum while empowering patients to become active protagonists in their care [21]. In 2017, the American Academy of Pediatrics (AAP) published a guide on pediatric integrative medicine, acknowledging the increased use of complementary therapies in this age group [22]. The guide reviewed evidence supporting therapies such as acupuncture, aromatherapy, and reflexology. Acupuncture was highlighted for its efficacy in reducing pain, nausea, and anxiety, particularly in oncology settings. Aromatherapy was recognized for its potential to enhance psychological well-being and alleviate symptoms like stress and fatigue through the use of essential oils. Reflexology, though supported by more limited evidence, was noted for its benefits in promoting relaxation and improving overall comfort. The AAP emphasized the importance of using these therapies alongside conventional treatments, recommending their application in clinical practice when delivered by certified practitioners and supported by emerging evidence [23].

Integrative oncology units are increasingly spreading worldwide, responding to the demand for complementary treatments mostly in adult, but also in pediatric oncology centers [24,25]. This trend is evident not only in high-income countries but also in those classified as low- and middle-income, highlighting a universal desire to explore and integrate a variety of therapeutic approaches applicable in cancer care [19,26]. Prominent U.S. oncology centers offer a range of complementary treatments including nutritional evaluation and advice, acupuncture, mind-body therapies, meditation, mindfulness, massage, reflexology, yoga, tai chi, music therapy, art therapy, aromatherapy, and physical training programs [27,28].

Currently, the Spanish public health system does not cover complementary integrative therapies, leaving patients to bear the cost of these interventions. To address this gap in our region, the Pediatric Integrative Oncology Unit (UOPI, for its abbreviation in Spanish: Unidad de Oncología Pediátrica Integrativa) was established in September 2019 within the oncology area of the Hospital Sant Joan de Déu, currently the SJD Pediatric Cancer Center Barcelona. This unit, the first of its kind in Spain, was made possible through philanthropic support, enabling the incorporation of complementary treatments into all oncologic patients’ therapeutic plans at no additional cost. The UOPI treats pediatric patients as well as adolescents and young adults, who may present with developmental cancers. Examples of these cancers include neuroblastoma, medulloblastoma, osteosarcoma, and germ cell tumors.

By offering complementary therapies integrated into clinical care, the UOPI seeks to enhance the overall treatment experience and address the multidimensional needs of pediatric cancer patients. This innovative approach reflects broader global trends and has the potential to serve as a model for expanding access to integrative oncology within national healthcare systems.

The aim of this study is to describe the acceptance rate and clinical activity of complementary treatments offered to patients, including acupuncture, aromatherapy, and reflexology, corresponding to the first two years of implementation of the UOPI. Nutritional counseling is part of the UOPI, and although it was provided to all patients, it was not evaluated in this study, as it is considered standard of the care rather than a complementary intervention.

## 2. Materials and Methods

This study was approved by the Medical Research Ethics Committee of Hospital Sant Joan de Déu (approval code: PIC-13-22). The study report was prepared in accordance with the Strengthening the Reporting of Observational Studies in Epidemiology (STROBE) guidelines [29] Appendix A.

The study was conducted at Sant Joan de Déu Children’s Hospital in Barcelona, in full compliance with local legislation and institutional regulations. Patient health data were retrospectively collected from clinical records. Since many patients no longer had active follow-ups at the hospital, obtaining specific informed consent beyond the general consent provided by oncology patients during their treatment was considered unfeasible. Consequently, a request for consent exemption was submitted, along with a data management commitment document signed by the principal investigator. This was accomplished in accordance with Organic Law 3/2018 of 5 December, on the Protection of Personal Data and the Guarantee of Digital Rights, ensuring that patient privacy and data security were maintained throughout the study.

### 2.1. Study Design and Setting

This retrospective, descriptive, observational study was conducted at the UOPI within the Hospital Sant Joan de Déu in Barcelona/Pediatric Cancer Center Barcelona, Spain, over a two-year period. The study took place in a single-center setting, focusing on the integrative oncology services provided by the UOPI.

The UOPI provides individualized care to all patients, beginning with personalized nutritional counseling as part of the broader therapeutic plan. Based on their clinical condition, treatment-associated toxicities, or specific therapeutic needs, patients are referred to the UOPI by their oncologist for evaluation and treatment. This may involve a variety of complementary therapies, including acupuncture (body acupuncture, auriculotherapy, moxibustion), Transcutaneous Electrical Nerve Stimulation (TENS), Transcutaneous Auricular Vagus Nerve Stimulation (taVNS), electroacupuncture, stiperpuncture, cross tape application, aromatherapy, and reflexology. These interventions are delivered during hospitalization to help alleviate symptoms and improve the overall well-being of the patients, ensuring an integrative, supportive approach throughout their cancer treatment journey.

Consultations regarding complementary therapies can be initiated at any stage of the patient’s illness, whether by healthcare providers, patients themselves, or their families. The UOPI offers a safe and relaxed environment for these discussions, with close and individualized follow-up to monitor and adjust care based on the patient’s evolving needs. The multidisciplinary team at the UOPI includes experienced healthcare professionals with over a decade of expertise in complementary treatments for both pediatric and adult patients. This team consists of a pediatrician with additional training in acupuncture, nutrition, and aromatherapy, one nutritionist, and four reflexologists, ensuring a broad spectrum of complementary care tailored to the patients’ needs. The team provides services across multiple hospital settings, including inpatient wards, outpatient clinics, daycare hospitals, the hematopoietic progenitor transplant unit, and intensive care units. In addition, cancer survivors have access to these treatments to manage late-stage toxicities arising from their previous cancer treatments.

### 2.2. Participants

The study included all oncologic patients evaluated by the Pediatric Integrative Oncology Unit (UOPI) between its establishment on 1 September 2019 and 30 September 2021; no additional exclusion or inclusion criteria were settled. The UOPI serves both pediatric patients and young adults who are transitioning from pediatric oncology care. As a result, the study sample comprised both pediatric and adult patients.

### 2.3. Data Collection

Data were collected retrospectively by three researchers who reviewed the patients’ electronic medical records. Patient confidentiality was strictly maintained throughout the data collection and analysis process in accordance with established ethical standards and institutional guidelines.

First, the researchers accessed patients’ electronic medical records in a pseudonymous manner, meaning that all identifiable information, such as names, medical record numbers, and other personal identifiers, was replaced with unique study codes. This ensured that the data could not be linked back to individual patients. Furthermore, the data collection form used for recording information was designed to exclude any identifying details, focusing solely on anonymized sociodemographic, clinical, and treatment-related variables. All data were stored in secure, password-protected files, accessible only to the research team.

A standardized data collection form was created using Microsoft Excel to ensure consistency. The extracted data included sociodemographic information (age at the time of appointment, gender, race, ethnicity, country of origin, and cancer type), hospital admission area (inpatients, outpatients, daycare hospital, transplant unit, special techniques areas, and intensive care unit), reasons for consultation, and details of the complementary treatment (type, number of sessions, and instances of rejection). A fourth researcher (S.P.J.) checked the accuracy of the data and reviewed the records for any mistakes, ensuring the reliability of the collected information.

No strategies for handling missing data were necessary, as there were no instances of missing data in the dataset. The completeness of the records ensured that all relevant information was captured and analyzed without the need for imputation or other data handling techniques.

### 2.4. Statistical Analysis

The demographic and clinical characteristics of the study population were summarized using SPSS software (version 29). Qualitative variables were described using absolute frequencies and percentages, while quantitative variables were summarized using the median, minimum, and maximum values.

## 3. Results

During the two-year study period, 433 pediatric oncology patients were visited in the UOPI.

### 3.1. Sociodemographic Data of Patients

The sociodemographic data of patients is summarized in Table 1. The median age at the first visit to the UOPI was 9 years (range: 0 to 34 years). Of the patients who attended, 266 were boys (61.4%) and 167 were girls (38.6%). Most patients (90.1%) were in active cancer treatment at the time of their UOPI visits, with patients receiving care at various stages of their disease across different care areas. Additionally, 7.6% of the patients were out of active therapy, while 2.3% were at the end-of-life care.

### 3.2. Hospital Areas and Reasons for Consultation

Participants were primarily visited in the ward (362), followed by the daycare hospital (107), a specialized unit within the hospital where patients receive care for procedures or treatments that do not require overnight hospitalization, outpatient clinic (87), stem cell transplant unit (7), procedures area (6), and intensive care unit (5) (Table 2). Some patients were visited in multiple areas of the center depending on their clinical status and needs.

Detailed information regarding the reasons for consultation was only available for acupuncture and aromatherapy. For all other techniques, the consultation reason was not systematically recorded. With respect to acupuncture and aromatherapy, the most common reasons for consultation were chemotherapy-induced nausea and vomiting, gastrointestinal motility disorders, pain, and stress management. Some patients presented with more than one reason for consultation. The remaining diagnoses, along with the complementary treatments received, are detailed in Table 3.

### 3.3. Provided Treatments

A detailed summary of the interventions provided by the UOPI during the study period is shown in Table 4.

#### 3.3.1. Acupuncture

Acupuncture was recommended to 227 cancer patients, with 215 (94.7%) accepting the treatment. The rejection rate was 5.3%, with nine parents (4.0%) and three patients (1.3%) declining the treatment. A total of 1,352 acupuncture sessions were performed over the two-year study period, with a median of 4.5 sessions per patient (range: 1 to 43 sessions). In addition to filiform needle acupuncture, non-insertional techniques were also used. These included cross tape, an adhesive tape applied to acupuncture points or areas of muscle tension; stiperpuncture, which involves placing small silicon-based tablets on acupuncture points; and moxibustion. Additionally, Transcutaneous Electrical Nerve Stimulation of acupuncture points and Transcutaneous Auricular Vagus Nerve Stimulation (taVNS) at ear acupuncture points, which were implemented in the last three months of the study period (Table 5).

#### 3.3.2. Aromatherapy

Aromatherapy treatments were recommended to 114 patients, all of whom (100%) accepted and participated in a single information session. This session provided specific instructions on using aromatherapy to alleviate cancer or cancer treatment-related toxicities such as nausea, pain, and anxiety.

#### 3.3.3. Reflexology

Reflexology was recommended to 134 participants, with 129 (96.1%) accepting the treatment. Despite restrictions imposed by SARS-CoV-2, the reflexology team conducted a total of 414 sessions over the two-year period.

## 4. Discussion

Integrative medicine combines conventional treatments with complementary therapies to holistically address patients’ physical and emotional needs. In pediatric oncology, this approach supports symptom management and enhances quality of life, which many families increasingly seek [30]. The Integrative Pediatric Oncology Unit at the Pediatric Cancer Center Barcelona was designed with this goal, offering therapies like acupuncture, aromatherapy, and reflexology as complements to standard care.

The findings of our study demonstrate a high level of acceptance and feasibility of implementing an Integrative Pediatric Oncology Unit within a comprehensive care model in a pediatric cancer center, with a global rejection rate of only 4.77%. Regarding reasons for consultation, acupuncture and aromatherapy were predominantly used for managing chemotherapy-induced nausea and vomiting, gastrointestinal motility disorders, pain (including neuropathic and immunotherapy-related pain), and stress or anxiety. The use of these therapies demonstrated adaptability to a broad spectrum of symptoms, reflecting their versatility in addressing both physical and psychological dimensions of care. Acupuncture was the most widely offered therapy (52.42% of patients) and had a high acceptance rate (94.7%), followed by aromatherapy, which had a perfect acceptance rate of 100%. This suggests that these interventions were particularly well-received for symptom relief and stress management.

To our knowledge, this is one of the first studies assessing the acceptability of complementary therapies within a pediatric oncology setting. Such low rejection rates highlight a growing willingness among families to explore integrative approaches, particularly when these therapies are seamlessly embedded into the care structure by trained professionals, reducing concerns about safety, accessibility, or additional costs.

A previous study conducted by Lim et al. in Singapore in 2006 assessed the prevalence of complementary and alternative medicine (CAM) use among pediatric cancer patients, reporting that 67.1% of patients had used at least one type of therapy [31]. However, a notable distinction of our study is that the complementary therapies were offered and administered within the same hospital by experienced healthcare professionals at no additional cost to the patients. This integrated approach is likely a key factor contributing to our study’s higher acceptance rate. As emphasized in previous studies, healthcare service integration and professional competence are crucial determinants of patient satisfaction, underscoring the value of embedding these services within hospital care models to enhance trust and accessibility [32].

Similarly, a 2017 study conducted at Columbia University Medical Center in the United States reported a 54% acceptance rate of acupuncture among 90 acupuncture-naïve children receiving cancer treatment [33]. In contrast, our study observed a much higher acceptance rate of 94.7%. This discrepancy could be attributed to the use of various acupuncture strategies in our setting, such as painless techniques with minimal puncture using semi-permanent needles, as well as non-invasive methods like cross tape, stiperpuncture, moxibustion, and transcutaneous electrical nerve stimulation (TENS). Additionally, the acupuncture sessions in our study were provided by a senior pediatrician acupuncturist, which may have contributed to a higher level of trust and confidence among patients and their families, leading to greater acceptance of the treatment. Evidence suggests that provider expertise and interpersonal communication significantly enhance patient satisfaction, particularly in sensitive populations such as pediatric patients [32]. These findings highlight the importance of tailoring complementary therapies to the pediatric population, ensuring both physical comfort and emotional reassurance through approaches that minimize invasiveness and build trust.

Lim et al.’s study also found that 55.1% of parents had not discussed their CAM usage with their child’s physician, which poses potential risks due to a lack of coordinated care [31]. The UOPI model directly addresses this issue by providing CAM services through highly trained healthcare providers who work closely with oncologists and are part of the multidisciplinary team. This collaboration enhances patient safety and satisfaction, as it ensures that complementary therapies are integrated into the patient’s overall treatment plan. The implementation of similar models of integrative care in the United States has demonstrated positive outcomes, which served as an inspiration for our own unit [7,34]. By fostering open communication and collaboration among healthcare providers, integrative units like the UOPI can mitigate risks associated with unregulated complementary therapy use, such as potential interactions with conventional treatments, delayed essential medical interventions, and exposure to unsafe practices [35], while promoting a more cohesive approach to care.

Despite the positive results, this study has several limitations. First, the COVID-19 pandemic imposed significant restrictions on our ability to conduct in-person visits. Reflexology sessions were suspended, and acupuncture interventions were limited to those deemed strictly necessary to minimize the risk of patient exposure to the virus. These changes likely impacted the patient care data we collected and may not accurately reflect service utilization in a post-pandemic era. Second, the retrospective nature of our study restricted the scope of factors that could be analyzed. Variables such as patients’ educational background, occupational status, type of employment, and symptom severity could not be assessed. Moreover, patient-reported outcomes such as satisfaction or perceived benefits of the therapies were not captured. Capturing these outcomes in future prospective studies will be essential to understand the full scope of benefits and identify areas for further improvement in service delivery.

The inclusion of young adults transitioning from pediatric oncology care also deserves attention. These patients, up to 34 years of age, may present with developmental cancers typically treated in pediatric settings. The inclusion of young adults may have influenced the acceptance rates, as the consent process differs for minors (who require parental consent) compared to adults (who provide independent consent). However, given the distribution of ages, the majority of patients were pediatric, with most participants falling under 18 years of age. Therefore, we do not consider this a significant concern for the interpretation of the results.

Moreover, while the acceptance rates of complementary therapies were high, the short and long-term benefits and possible adverse effects of these treatments remain areas that require further investigation in the pediatric oncologic population. Future research should prioritize longitudinal studies to evaluate the sustained impact of complementary therapies on clinical outcomes, quality of life, and psychosocial health, as well as potential interactions with conventional cancer treatments. Additionally, incorporating feedback from patients and families can provide invaluable insights into optimizing integrative oncology models for broader implementation.

## 5. Conclusions

The high acceptance rates of the UOPI’s complementary therapies, with over 94% for acupuncture and reflexology and 100% for aromatherapy, suggest that integrative oncology can be successfully incorporated into pediatric cancer care. These therapies were primarily offered in the ward, daycare hospital, and outpatient clinic, reflecting their integration into diverse care settings and clinical stages.

In addition to the high acceptance rates, this study highlights important findings regarding the reasons for consultation and types of interventions offered. Acupuncture and aromatherapy were most commonly used to address chemotherapy-induced nausea and vomiting, gastrointestinal motility disorders, pain management (including neuropathic, immunotherapy-related, and musculoskeletal pain), and stress/anxiety. Aromatherapy was also applied for conditions such as insomnia, dermatological issues, and respiratory symptoms. These findings demonstrate the broad applicability of these therapies for symptom management in pediatric oncology patients.

Future research should focus on the efficacy, safety, and long-term outcomes of these therapies, as well as identifying best practices for their implementation in various healthcare settings. Such studies will be crucial for establishing evidence-based guidelines and ensuring that complementary therapies are used effectively and safely in pediatric oncology. Ultimately, this approach may contribute to a more holistic, patient-centered model of care that better meets the needs of pediatric cancer patients and their families.

## Figures and Tables

**Table 1 cancers-17-00222-t001:** Demographic characteristics.

Characteristics	Number of Patients (n = 433)	%
**Age in first visit** (years)		
0–4	128	29.6
5–9	112	265.9
10–14	104	24.0
15–19	81	18.7
20–24	7	1.6
25–29	0	0
30–34	1	0.2
Median (Range)	9 (0 to 33)	
**Sex**		
Male	266	61.4
Female	167	38.6
**Race**		
Caucasian	414	95.6
Asian	13	3.0
Black	6	1.4
**Origin**		
Spain	375	86.6
Europe	25	5.8
Asia	23	5.3
Latin America	7	1.6
Africa	3	0.7
**Active treatment**		
Yes	390	90.0
No	43	10.0
**Out of active therapy**	33	7.6
**End of life situation**		
Yes	10	2.3
No	423	97.7

**Table 2 cancers-17-00222-t002:** Area of treatment.

Hospital Area	Number of Patients *
Ward	362
Outpatient clinic	87
Day care hospital	107
Stem Cell Transplant unit	7
Procedures room	6
Intensive care unit	5

* Several patients were seen in multiple hospital areas.

**Table 3 cancers-17-00222-t003:** Consultation reasons.

Condition *	Acupuncture	Aromatherapy
**Gastrointestinal motility**		
Chemotherapy-induced nausea/vomiting	113	
Constipation	59	5
Diarrhea	36	
**Pain**		
Abdominal pain	59	
Pain Immunotherapy related	4	
Headache	29	
Headache lumbar punction related	6	
Neuropathic pain	28	9
Arthralgia	18	
Phantom limb	4	24
Scar pain	12	
Musculoskeletal	49	
**Stress/Anxiety**	61	43
**Respiratory symptoms**		
Mucus	5	7
Cough	1	
**Night terrors**	13	
**Anal fissure**	6	
**Allergic reaction**	14	
**Insomnia**	29	
**Mucositis**		4
**Mutism**	4	
**Asthenia**	18	
**Loss of appetite**	19	
**Dermatological issues**		26
**Neurogenic bladder**	1	
**Hot flushes**	3	
**Tinnitus**	4	
**Bruxism**	3	

* Consultation reasons were not systematically reported for reflexology.

**Table 4 cancers-17-00222-t004:** Provided treatments and acceptance.

Treatment	Patients Offered n (%)	Patients that Accepted n (%)	Rejection of the Treatment n (%)	Number of Treatments Provided n
Acupuncture	227 (52.42)	215 (94.7)	12 (5.3)	1352
Aromatherapy	196 (45.2)	196 (100)	0	196 *
Reflexology	129 (29.7)	124 (96.1)	5 (3.9)	414

* Aromatherapy was a single session intervention.

**Table 5 cancers-17-00222-t005:** Description of the acupuncture treatment components.

Acupuncture Technique	Number of Participants that Received Acupuncture, n (%)
**Filiform needle**	215 (100)
**Non-insertional techniques**	62 (28.8)
Cross tape	10 (4.6)
Stipper	10 (4.6)
Moxibustion	21 (9.7)
Mixed	21 (9.7)
**Electroacupuncture**	28 (29.7)
TaVNS	19 (8.8)
TENS	3 (1.4)

TaVNS: Transcutaneous Auricular Vagus Nerve Stimulation; TENS: Transcutaneous Electrical Nerve Stimulation.

## Data Availability

The original data presented in the study are openly available at https://figshare.com/articles/dataset/Dataset/27118425?file=49445457 (access on 17 december 2024).

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
