# Peer review of "Patient Acceptability of the First Integrative Pediatric Oncology Unit in Spain—The Pediatric Cancer Center Barcelona Experience: A Retrospective Study"

_cancers, 2025, doi:10.3390/cancers17020222_

Round 1
Reviewer 1 Report
Comments and Suggestions for Authors
The authors present a retrospective analysis of the implementation and feasibility of an integrative pediatric oncology unit at the Pediatric Cancer Center Barcelona, Spain. The study highlights the acceptance of various complementary therapies among pediatric cancer patients and their families over a two-year period. The findings reveal high acceptance rates for acupuncture, aromatherapy, and reflexology, supporting the feasibility of such interventions in a patient-centered care model.
- The establishment of an integrative pediatric oncology unit is a novel and initiative, especially within the context of public healthcare systems like Spain’s, where such complementary therapies are not commonly covered.
- The study covers a significant sample size (433 patients) over two years, allowing for meaningful insights into patient demographics, therapy acceptance, and service utilization.
- The reported acceptance rates (94.7% for acupuncture, 100% for aromatherapy, and 96.1% for reflexology) underscore the receptiveness of patients and families to complementary therapies, reinforcing their potential role in supportive care.
- The study aligns with a holistic care model, aiming to improve the quality of life for pediatric oncology patients, which is particularly commendable in challenging clinical scenarios.
- The methods section could provide more detail about the selection criteria for patients, the process of recommending therapies, and the role of healthcare professionals in delivering these interventions.
- While the study highlights acceptance rates, it lacks robust clinical or quality-of-life outcome measures. Including metrics like patient-reported outcomes, symptom relief, or stress reduction would strengthen the findings.
Author Response
Comment 1: The methods section could provide more detail about the selection criteria for patients, the process of recommending therapies, and the role of healthcare professionals in delivering these interventions.
Response 1: Thank you for your insightful comment. We have now included more detailed information in the methods section regarding the patient selection criteria. lines 118-121 and 180-182.
Comment 2: While the study highlights acceptance rates, it lacks robust clinical or quality-of-life outcome measures. Including metrics like patient-reported outcomes, symptom relief, or stress reduction would strengthen the findings.
Response 2: Thank you for this valuable suggestion. We agree that incorporating clinical outcomes and patient-reported metrics would provide a more comprehensive understanding of the impact of complementary therapies. While our study focused primarily on the acceptance rates of the therapies, we acknowledge that symptom relief and quality of life are key aspects of evaluating their effectiveness. This is discussed in the limitations section.
Reviewer 2 Report
Comments and Suggestions for Authors
Comments to the authors – Martinez Garcia et al, 2024
First of all, I would like to congratulate the authors on their highly relevant and pioneer work, which constitutes a meaningful contribution to the advance of patient-centered and integrative care in pediatric oncology in European countries.
The publication is well written, easy to read and easy to follow, and meaningful at the regional and global level. It was a pleasure to review such publication, and I hope the team considers contributing other publications in related topics.
Below are very minor suggestions that I believe may improve the clarity of some of the points made in the publication. Feel free to contact me if you need any clarification. I am looking forward to seeing this publication available soon!
Warmly,
Reviewer
--------------------------------------------------------------------------------------------------------------
LINE 99: Please delete the word “alternative” as it refers to using a non-conventional therapy instead of a conventional treatment (not what you are trying to describe here)
Currently, the Spanish public health system does not cover complementary and 98
integrative medicine (CIM), leaving patients to bear the cost of these alternative therapies. 99
LINE 199-200: Just as a clarification, may want to either edit out or explain why ages up to 34 are included in a pediatric sample. Usually pediatric samples include ages up to age 18 or 21 then is considered “young adults” in the peer review literature.
The sociodemographic data of patients is summarized in Table 1. The median age at 199 the first visit to the UOPI was 9 years (range: 0 to 34 years). Of the patients who attended, 200
LINE 207: Briefly explain what is “daycare hospital” in this context (it may not be clear for readers of all countries).
Participants were primarily visited in the ward (362), followed by the daycare hospital 207 (107), outpatient clinic (87), stem cell transplant unit (7), procedures area (6), and in-208 tensive care unit (5) (Table 2). Some patients were visited in multiple areas of the center 209 depending on their clinical status and needs.
Table 3.
“N/V induced chemotherapy” (should read: chemotherapy-induced nausea/vomiting or chemo-induced nausea/vomiting)
LINE 231: SEE COMMENT BELOW
In addition to filiform needle acupuncture, various techniques were progressively intro-230 duced, including electroacupuncture and Transcutaneous Auricular Vagus Nerve 231 Stimulation (taVNS) in the last three months of the study period (Table 5). 232
Comment: Transcutaneous Auricular Vagus Nerve Stimulation (taVNS) is a technique in which a microcurrent is applied to the vagus nerve. This is not related in any form to acupuncture or electro-acupuncture (which is a technique in which specific acupuncture points are stimulated by an electrical current).
It may be helpful for the authors to describe in the methods section that, in addition to consistently offering acupuncture and aromatherapy, they also offered at some point of the study other modalities such as reflexology, Transcutaneous Auricular Vagus Nerve Stimulation (taVNS) and TENs. TENS is not even considered a complementary therapy in the US, given that is a conventional treatment in physical therapy (also referred as “kinesiology” in Spanish speaking countries).
It may also be helpful to very briefly describe (even if only a footnote) the non-insertion modalities used, as they are not all very well known in other countries, perhaps highlighting differences between techniques that may be available the US and European countries. I had to search for cross tape stimulation and stiperpuncture, as neither was familiar to me even when I am familiar with all non-insertion techniques used in the US and Latin America. For example, in the US, Japanese style acupuncture is considered the gentlest insertion technique, therefore the most common technique used in pediatrics and in palliative care along with laser acupuncture and tuning forks.
Author Response
Comment 1. LINE 99: Please delete the word “alternative” as it refers to using a non-conventional therapy instead of a conventional treatment (not what you are trying to describe here)
Response1. Thank you for the suggestion, this has been corrected
Comment 2. LINE 199-200: Just as a clarification, may want to either edit out or explain why ages up to 34 are included in a pediatric sample. Usually pediatric samples include ages up to age 18 or 21 then is considered “young adults” in the peer review literature.
Response 2. We appreciate your comment. To clarify, we have added the following information:
“The UOPI treats pediatric patients as well as adolescents and young adults, who may present with developmental cancers. Examples of these cancers include neuroblastoma, medulloblastoma, osteosarcoma, and germ cell tumors.” Line 118-121.
Additionally, we have included this information in the limitations section:
“The inclusion of young adults transitioning from pediatric oncology care also de-serves attention. These patients, up to 34 years of age, may present with developmental cancers typically treated in pediatric settings. The inclusion of young adults may have influenced the acceptance rates, as the consent process differs for minors (who require parental consent) compared to adults (who provide independent consent). However, given the distribution of ages, the majority of patients were pediatric, with most par-ticipants falling under 18 years of age. Therefore, we do not consider this a significant concern for the interpretation of the results.” Line 344-351.
Comment 3. LINE 207: Briefly explain what is “daycare hospital” in this context (it may not be clear for readers of all countries).
Response 3. Thank you for your suggestion. We have added a brief description of the "daycare hospital" for clarity:
“Participants were primarily visited in the ward (362), followed by the daycare hospital (107), a specialized unit within the hospital where patients receive care for procedures or treatments that do not require overnight hospitalization, outpatient clinic (87), stem cell transplant unit (7), procedures area (6), and intensive care unit (5) (Table 2). Some patients were visited in multiple areas of the center depending on their clinical status and needs.” Line 225-229
Comment 4. Table 3. “N/V induced chemotherapy” (should read: chemotherapy-induced nausea/vomiting or chemo-induced nausea/vomiting)
Response 4. We appreciate your comment. We have corrected the terminology in Table 3
Comment 5.: Transcutaneous Auricular Vagus Nerve Stimulation (taVNS) is a technique in which a microcurrent is applied to the vagus nerve. This is not related in any form to acupuncture or electro-acupuncture (which is a technique in which specific acupuncture points are stimulated by an electrical current).
It may be helpful for the authors to describe in the methods section that, in addition to consistently offering acupuncture and aromatherapy, they also offered at some point of the study other modalities such as reflexology, Transcutaneous Auricular Vagus Nerve Stimulation (taVNS) and TENs. TENS is not even considered a complementary therapy in the US, given that is a conventional treatment in physical therapy (also referred as “kinesiology” in Spanish speaking countries).
Response 5. Thank you for your observation. We have clarified the description of Transcutaneous Auricular Vagus Nerve Stimulation (taVNS) and Transcutaneous Electrical Nerve Stimulation (TENS) in the methods section.
“Additionally, Transcutaneous Electrical Nerve Stimulation of acupuncture points and Transcutaneous Auricular Vagus Nerve Stimulation (taVNS) at ear acupuncture points, which were implemented in the last three months of the study period (Table 5).” Line 251-255.
Comment 6. It may also be helpful to very briefly describe (even if only a footnote) the non-insertion modalities used, as they are not all very well known in other countries, perhaps highlighting differences between techniques that may be available the US and European countries. I had to search for cross tape stimulation and stiperpuncture, as neither was familiar to me even when I am familiar with all non-insertion techniques used in the US and Latin America. For example, in the US, Japanese style acupuncture is considered the gentlest insertion technique, therefore the most common technique used in pediatrics and in palliative care along with laser acupuncture and tuning forks.
Response 6. Thank you for your suggestion to clarify the non-insertional modalities. We have added brief descriptions of these techniques in the revised manuscript:
“Acupuncture was recommended to 227 cancer patients, with 215 (94.7%) accepting the treatment. The rejection rate was 5.3%, with 9 parents (4.0%) and 3 patients (1.3%) declining the treatment. A total of 1,352 acupuncture sessions were performed over the two-year study period, with a median of 4.5 sessions per patient (range: 1 to 43 sessions). In addition to filiform needle acupuncture, non-insertional techniques were also used. These included cross tape, an adhesive tape applied to acupuncture points or areas of muscle tension; stiperpuncture, which involves placing small silicon-based tablets on acupuncture points; and moxibustion. Additionally, various techniques were progres-sively introduced, including electroacupuncture Transcutaneous Electrical Nerve Stimulation of acupuncture points and Transcutaneous Auricular Vagus Nerve Stimu-lation (taVNS) at ear acupuncture points, which were implemented in the last three months of the study period (Table 5).” Line 244-255
Reviewer 3 Report
Comments and Suggestions for Authors
This retrospective study provides interesting results about the acceptance rate of complementary therapies such as acupuncture, aromatherapy, and reflexology offered by a multidisciplinary team to pediatric patients. This investigation adds complementary findings to the studies testing the efficacy of such interventions, which is not the unique factor that must be considered when implementing interventions. The manuscript is well-written and the study is clearly presented. However, I have some suggestions to improve the manuscript’s contents.
Introduction
The introduction is clear and well-structured. However, I suggest the authors add a paragraph describing acupuncture, aromatherapy, reflexology, data supporting their efficacy, and guidelines supporting their use in clinical practice.
Further, the conceptual definition of complementary therapies should be provided. It is crucial to clarify the type of intervention under investigation and what definition we are referring to. The authors use both the terms "complementary" and "integrative" throughout the text, which are two different concepts. Please be consistent
Methods
I suggest the authors add a statement about eligibility criteria. The authors stated that they included "all oncologic patients evaluated by the Pediatric Integrative Oncology Unit (UOPI). Please specify the age range and if there are any additional inclusion or exclusion criteria. Otherwise, they can say, for example, "no additional exclusion or inclusion criteria were settled". This clarification guarantees that the reporting is complete.
Further, the authors included patients who had consultations with the UOPI team. Therefore, the focus is on pediatric patients. However, in the results section, the sample's age range was 0-34 years. Please provide a clarification about this inconsistency. What is the age range for defining the pediatric population? This aspect is also crucial for interpreting the study's results as adults (≥ 18 years) provide their consent independently, whereas in the younger group, the parents provide the consent. This difference could influence the acceptance rates. I think this is also a considerable study limitation since it completely changes the focus on the target population in relation to the results.
Discussion
Please add a referenced sentence that supports this statement: "However, a notable distinction of our study is that the complementary therapies were offered and administered within the same hospital by experienced healthcare professionals at no additional cost to the patients. This integrated approach is likely a key factor contributing to our study's higher acceptance rate".
This sentence should be referenced: "Additionally, the acupuncture sessions in our study were provided by a senior paediatrician acupuncturist, which may have contributed to a higher level of trust and confidence among patients and their families, leading to greater acceptance of the treatment". How can you say that a senior paediatrician can contribute to a greater acceptance rate?
Lines 290-292: The authors mentioned risks related to complementary therapies. I suggest to expand on this aspect.
I suggest the authors add in the limitation section that due to the study's retrospective design, it was impossible to analyze additional factors that could influence the study's results, such as educational background, occupational status, type of employment, symptoms severity and so on.
Overall, the study's results should be more widely discussed in light of the age range and symptomatological outcomes that required the interventions. In addition, there are additional relevant results that were not discussed.
Conclusion
I understand that the main outcome was the acceptance rate, but this study brings to light further relevant findings. Therefore, additional conclusions should be briefly reported, including the main reasons for consultation and the type of intervention mostly offered in relation to the acceptance rate and symptoms.
Author Response
Comment 1: The introduction is clear and well-structured. However, I suggest the authors add a paragraph describing acupuncture, aromatherapy, reflexology, data supporting their efficacy, and guidelines supporting their use in clinical practice. Further, the conceptual definition of complementary therapies should be provided. It is crucial to clarify the type of intervention under investigation and what definition we are referring to. The authors use both the terms "complementary" and "integrative" throughout the text, which are two different concepts. Please be consistent.
Response 1:
We appreciate your insightful suggestion. In response, we have added a brief description of the interventions in the revised manuscript:
“The guide reviewed evidence supporting therapies such as acupuncture, aromatherapy, and reflexology. Acupuncture was highlighted for its efficacy in reducing pain, nausea, and anxiety, particularly in oncology settings. Aromatherapy was recognized for its potential to enhance psychological well-being and alleviate symptoms like stress and fatigue through the use of essential oils. Reflexology, though supported by more limited evidence, was noted for its benefits in promoting relaxation and improving overall comfort. The AAP emphasized the importance of using these therapies alongside conventional treatments, recommending their application in clinical practice when delivered by certified practitioners and supported by emerging evidence [21].” Lines 91-101.
Furthermore, we have added definitions for complementary and integrative therapies, ensuring consistency in their usage throughout the document:
“Complementary therapies are defined as medical practices and products not typically part of conventional medical care but used alongside standard treatments to improve patient well-being and quality of life. Integrative medicine, on the other hand, combines complementary therapies with conventional medical treatments in a coordinated manner, focusing on a holistic, patient-centered approach to care [16].” Lines 81-85.
Comment 2: I suggest the authors add a statement about eligibility criteria. The authors stated that they included "all oncologic patients evaluated by the Pediatric Integrative Oncology Unit (UOPI)." Please specify the age range and if there are any additional inclusion or exclusion criteria. Otherwise, they can say, for example, "no additional exclusion or inclusion criteria were settled." This clarification guarantees that the reporting is complete.
Further, the authors included patients who had consultations with the UOPI team. Therefore, the focus is on pediatric patients. However, in the results section, the sample's age range was 0-34 years. Please provide clarification about this inconsistency. What is the age range for defining the pediatric population? This aspect is also crucial for interpreting the study's results as adults (≥ 18 years) provide their consent independently, whereas in the younger group, the parents provide the consent. This difference could influence the acceptance rates. I think this is also a considerable study limitation since it completely changes the focus on the target population in relation to the results.
Response 2:
Thank you for pointing this out. We have clarified the inclusion criteria in the manuscript:
“The study included all oncologic patients evaluated by the Pediatric Integrative Oncology Unit (UOPI) between its establishment on September 1, 2019, and September 30, 2021; no additional exclusion or inclusion criteria were settled. The UOPI serves both pediatric patients and young adults who are transitioning from pediatric oncology care. As a result, the study sample comprised both pediatric and adult patients.” Lines 177-182.
Additionally, we have clarified that the pediatric unit includes both children and young adults:
“The UOPI treats pediatric patients as well as adolescents and young adults, who may present with developmental cancers. Examples of these cancers include neuroblastoma, medulloblastoma, osteosarcoma, and germ cell tumors.” Lines 118-121.
“The inclusion of young adults transitioning from pediatric oncology care also deserves attention. These patients, up to 34 years of age, may present with developmental cancers typically treated in pediatric settings. The inclusion of young adults may have influenced the acceptance rates, as the consent process differs for minors (who require parental consent) compared to adults (who provide independent consent). However, given the distribution of ages, the majority of patients were pediatric, with most participants falling under 18 years of age. Therefore, we do not consider this a significant concern for the interpretation of the results.” Lines 344-351.
Comment 3: Please add a referenced sentence that supports this statement: "However, a notable distinction of our study is that the complementary therapies were offered and administered within the same hospital by experienced healthcare professionals at no additional cost to the patients. This integrated approach is likely a key factor contributing to our study's higher acceptance rate."
Response 3:
We appreciate your suggestion. In response, we have added a referenced sentence to support the statement:
“However, a notable distinction of our study is that the complementary integrative therapies were offered and administered within the same hospital by experienced healthcare professionals at no additional cost to the patients. This integrated approach is likely a key factor contributing to our study's higher acceptance rate. As emphasized in previous studies, healthcare service integration and professional competence are crucial determinants of patient satisfaction, underscoring the value of embedding these services within hospital care models to enhance trust and accessibility [30].” Lines 296-303.
Comment 4: This sentence should be referenced: "Additionally, the acupuncture sessions in our study were provided by a senior pediatrician acupuncturist, which may have contributed to a higher level of trust and confidence among patients and their families, leading to greater acceptance of the treatment." How can you say that a senior pediatrician can contribute to a greater acceptance rate?
Response 4:
We have addressed this concern by referencing evidence that highlights the importance of provider expertise in improving patient satisfaction:
“Evidence suggests that provider expertise and interpersonal communication significantly enhance patient satisfaction, particularly in sensitive populations such as pediatric patients [30].” Lines 313-316.
Comment 5: Lines 290-292: The authors mentioned risks related to complementary therapies. I suggest expanding on this aspect.
Response 5:
We have expanded this section to provide further clarification on the risks:
“By fostering open communication and collaboration among healthcare providers, integrative units like the UOPI can mitigate risks associated with unregulated complementary therapy use, such as potential interactions with conventional treatments, delayed essential medical interventions, and exposure to unsafe practices [33].” Lines 327-330.
Comment 6: I suggest the authors add in the limitation section that due to the study's retrospective design, it was impossible to analyze additional factors that could influence the study's results, such as educational background, occupational status, type of employment, symptoms severity, and so on.
Response 6:
We have added this information in the limitations section to clarify the scope of the study:
“Despite the positive results, this study has several limitations. First, the COVID-19 pandemic imposed significant restrictions on our ability to conduct in-person visits. Reflexology sessions were suspended, and acupuncture interventions were limited to those deemed strictly necessary to minimize the risk of patient exposure to the virus. These changes likely impacted the patient care data we collected and may not accurately reflect service utilization in a post-pandemic era. Second, due to the retrospective nature of our study, the scope of factors that could be analyzed was restricted. Variables such as patients' educational background, occupational status, type of employment, and symptom severity could not be assessed. Moreover, patient-reported outcomes such as satisfaction or perceived benefits of the therapies were not captured. Capturing these outcomes in future prospective studies will be essential to understand the full scope of benefits and identify areas for further improvement in service delivery.” Lines 332-343.
Comment 7: Overall, the study's results should be more widely discussed in light of the age range and symptomatological outcomes that required the interventions. In addition, there are additional relevant results that were not discussed.
Response 7:
We have expanded the discussion of our study's results to address the broader context, including age range and symptomatology:
“The findings of our study demonstrate a high level of acceptance and feasibility of implementing an Integrative Pediatric Oncology Unit within a comprehensive care model in a pediatric cancer center, with a global rejection rate of only 4.77%. Regarding reasons for consultation, acupuncture and aromatherapy were predominantly used for managing chemotherapy-induced nausea and vomiting, gastrointestinal motility disorders, pain (including neuropathic and immunotherapy-related pain), and stress or anxiety. The use of these therapies demonstrated adaptability to a broad spectrum of symptoms, reflecting their versatility in addressing both physical and psychological dimensions of care. Acupuncture was the most widely offered therapy (52.42% of patients) and had a high acceptance rate (94.7%), followed by aromatherapy, which had a perfect acceptance rate of 100%. This suggests that these interventions were particularly well-received for symptom relief and stress management.” Lines 277-288.
Comment 8: Conclusion. I understand that the main outcome was the acceptance rate, but this study brings to light further relevant findings. Therefore, additional conclusions should be briefly reported, including the main reasons for consultation and the type of intervention mostly offered in relation to the acceptance rate and symptoms.
Response 8:
We have addressed this in the conclusion section to highlight the broader findings of the study:
“The high acceptance rates of the UOPI's complementary therapies, with over 94% for acupuncture and reflexology and 100% for aromatherapy, suggest that integrative oncology can be successfully incorporated into pediatric cancer care. These therapies were primarily offered in the ward, daycare hospital, and outpatient clinic, reflecting their integration into diverse care settings and clinical stages. In addition to the high acceptance rates, this study
Round 2
Reviewer 3 Report
Comments and Suggestions for Authors
I thank the authors for carefully and punctually addressing all my comments and suggestions. There is just a comment that has not been addressed. The authors added a definition of complementary therapies as suggested. However, as complementary and integrative therapies are defined differently in the literature; therefore, I recommend the authors be consistent with the terms throughout the manuscript by choosing one of these approaches starting from the title to the conclusions.
Author Response
Comment 1:
I thank the authors for carefully and punctually addressing all my comments and suggestions. There is just a comment that has not been addressed. The authors added a definition of complementary therapies as suggested. However, as complementary and integrative therapies are defined differently in the literature; therefore, I recommend the authors be consistent with the terms throughout the manuscript by choosing one of these approaches starting from the title to the conclusions.
Response 1:
Thank you for highlighting the importance of consistent terminology throughout the manuscript. We agree that "complementary" and "integrative" are distinct yet sometimes used interchangeably in the literature. Following your suggestion, we have revised the manuscript to ensure consistency in the use of these terms.
We now use the term "complementary" specifically to refer to the therapies or interventions that are used alongside conventional medical treatments to manage symptoms, improve quality of life, or support overall patient well-being. The term "integrative" is reserved for describing the unit or care model that integrates these complementary therapies with standard medical care, reflecting a coordinated, holistic approach.
This distinction aligns with the definitions provided by the National Center for Complementary and Integrative Health (NCCIH), which uses "complementary" to describe therapies and "integrative" to refer to care that combines complementary therapies with conventional care. Further details can be found at NCCIH’s website (https://www.nccih.nih.gov/health/complementary-alternative-or-integrative-health-whats-in-a-name).
We have carefully reviewed the manuscript to address all previous inconsistencies in terminology. However, we would be pleased to receive further feedback if there are any instances we may have missed. Your input is invaluable in helping us ensure clarity and precision throughout the text.